# Nutrition in Gastrointestinal Disease: Liver, Pancreatic, and Inflammatory Bowel Disease

**DOI:** 10.3390/jcm8081098

**Published:** 2019-07-25

**Authors:** Lena J. Storck, Reinhard Imoberdorf, Peter E. Ballmer

**Affiliations:** 1Kantonsspital Winterthur, Department of Medicine, 8401 Winterthur, Switzerland; 2Zürcher RehaZentrum Davos, 7272 Davos Clavadel, Switzerland

**Keywords:** monitoring, malnutrition, micronutrient deficiency, inflammation, oral nutritional supplements, artificial nutrition

## Abstract

Liver, pancreatic, and inflammatory bowel diseases are often associated with nutritional difficulties and necessitate an adequate nutritional therapy in order to support the medical treatment. As most patients with non-alcoholic fatty liver disease are overweight or obese, guidelines recommend weight loss and physical activity to improve liver enzymes and avoid liver cirrhosis. In contrast, patients with alcoholic steatohepatitis or liver cirrhosis have a substantial risk for protein depletion, trace elements deficiency, and thus malnutrition. Patients with chronic pancreatitis and patients with inflammatory bowel disease have a similar risk for malnutrition. Therefore, it clearly is important to screen these patients for malnutrition with established tools and initiate adequate nutritional therapy. If energy and protein intake are insufficient with regular meals, oral nutritional supplements or artificial nutrition, i.e., tube feeding or parenteral nutrition, should be used to avoid or treat malnutrition. However, the oral route should be preferred over enteral or parenteral nutrition. Acute liver failure and acute pancreatitis are emergencies, which require close monitoring for the treatment of metabolic disturbances. In most patients, energy and protein requirements are increased. In acute pancreatitis, the former recommendation of fasting is obsolete. Each disease is discussed in this manuscript and special recommendations are given according to the pathophysiology and clinical routine.

## 1. Introduction

Gastrointestinal diseases are often associated with nutritional deficiencies. The complications range from digestive problems to nutrient absorption disorders and necessitate an adequate nutritional therapy in order to support the medical treatment. In this article, we focus on diseases of three important organs: Liver, pancreas, and intestine.

The liver is the main metabolic organ of the human organism because of its multiple functions; e.g., the liver controls the glucose homeostasis by regulation of glycogen synthesis, glycogenolysis, and gluconeogenesis. In addition, important body proteins and lipids are metabolized and synthesized in liver cells. Next to these metabolic pathways, the liver has an exocrine function by secretion of bile acids and detoxification of ammonia by urea production and glutamine synthesis. Taken these extensive functions into account, it is obvious that liver diseases and thus restrictions in liver functions have far-reaching consequences also for clinical nutrition. An important and by dietary habits increasing liver disease is non-alcoholic fatty liver disease (NAFLD), when the fat content of hepatocytes increases [1,2]. NAFLD aggregates benign hepatosteatosis, progressive and inflammatory non-alcoholic steatohepatitis (NASH) [3]. Insulin resistance, hyperinsulinemia, elevated plasma free fatty acids, fatty liver, hepatocyte injury, liver inflammation, oxidative stress, mitochondrial dysfunction, imbalanced pro-inflammatory cytokines, and fibrosis characterize NAFLD [4]. The other main cause for the development of steatosis is alcohol abuse. Chronic high alcohol consumption causes reduced fatty-acid oxidation as well as increased triglyceride synthesis and deposition, and thereby supports the development of alcoholic steatohepatitis (ASH). Usually, steatohepatitis can be reversible if the relevant noxious agent is eliminated. Inadequate or lack of treatment over many years increases the risk for the development of a liver cirrhosis. Thus, 15–20% of patients with NAFLD will develop liver cirrhosis [5]. Next to these liver diseases that develop over many years, acute liver failure may occur usually as a medical emergency.

Another important organ, which can develop inflammatory processes, is the pancreas. This organ plays a central role in digestion due to its exocrine and endocrine function. A total of 98% of pancreatic tissue is part of the exocrine function, whereby pancreatic enzymes are secreted for food digestion. The endocrine part is located in the Langerhans cells, secreting insulin and glucagon for regulation of glucose homeostasis. Acute pancreatitis is a severe disease, causing self-digestion of the pancreas due to prematurely activated digestive enzymes. Patients with chronic pancreatitis have recurrent inflammatory episodes that replace the pancreatic parenchyma by fibrous connective tissue resulting in a progressive loss of exocrine and endocrine function. Characteristic complications and symptoms are pain, pseudocysts, and pancreatic duct stenosis [6].

The largest digestive organ is the intestine, which regulates the absorption of nutrients. Two important diseases of the intestine are Crohn’s disease (CD) and ulcerative colitis (UC) that are characterized by periods of remission and inflammatory flare-ups [7]. In CD, the small and large intestine as well as the mouth, esophagus, stomach, and the anus can be affected with typical ulcerations that occur discontinuously. In contrast, UC mostly affects the colon and rectum and usually shows a continuous pattern of the mucosa. Despite multiple differences, these two diseases share similar symptoms such as abdominal pain, diarrhea, and malnutrition. Hence, these diseases were grouped in the term inflammatory bowel disease (IBD) [7,8].

Nutritional care is clearly important in the treatment of patients with IBD, pancreas or liver disease. Nutritional management includes prevention and/or treatment of malnutrition and micronutrient deficiencies as well as specific recommendations for each condition.

## 2. Liver Diseases

### 2.1. Non-Alcoholic Fatty Liver Disease (NAFLD)

So far, there are no approved pharmacological therapies for NAFLD [3]. One reason is that the pathophysiology of NAFLD is not yet fully understood despite enormous advance in this field of research. Since overnutrition is the key problem of NAFLD and most of the patients are overweight or obese, weight loss is an obvious therapeutic possibility, hence, intensive lifestyle interventions are well studied [4,5]. The guideline on clinical nutrition in liver disease recommend a 7–10% weight loss to improve steatosis and liver enzymes [3,9]. To improve fibrosis, a weight loss of more than 10% is necessary [10]. Lifestyle interventions including diet and physical activity should be the first-line treatment and only if all efforts fail, bariatric surgery should be proposed [11,12].

In addition, the composition of the diet may also have an effect on liver fat. Low-carbohydrate diet may be helpful with weight loss, but over a long time, a low-carbohydrate diet stimulated NAFLD pathogenesis in an animal model [4]. It is speculated that the carbohydrate’s composition is crucial [4]. For example, fructose can easily induce metabolic complications in the liver and in contrast, fiber might be helpful to maintain blood glucose and thus prevent NAFLD. Furthermore, a fat-rich diet induces hepatic steatosis, but only saturated fatty acids are detrimental for the liver metabolism. Monounsaturated fatty acids might be beneficial and polyunsaturated fatty acids might even be a treatment option for NAFLD [4].

The evidence suggest that increased oxidative stress and changes in several molecular factors like pro- and anti-inflammatory cytokines are mainly involved in the progression of NAFLD [13]. Therefore, antioxidants like vitamin C or polyphenols (e.g., resveratrol, curcumin, quercetin, anthocyanin, green tea polyphenols) might have beneficial effects to improve NAFLD [4]. Kitade et al. (2017) showed that the dietary administration of the carotenoids β-crypoxanthin and astaxanthin not only prevents but also reverses NASH progression in mice by regulating M1/M2 macrophage/Kupffer cell polarization [13]. NAFLD induced by diets high in sucrose/fructose or fat can be prevented or improved by soy protein (β-Conglycinin) by decreasing the expression and function of the two nuclear receptors SREBP-1c and PPAR γ2. Fish oil with ω-3 fatty acids inhibits SREBP-1c activity, but controversially increases PPAR γ2 expression [13,14]. In summary, research results for the effects of micronutrients on NAFLD showed positive effects on some factors in liver metabolism. However, it remains unclear whether the use of antioxidants and ω-3 fatty acids improve liver disease, thus, further investigations are needed and no recommendation can be given [4]. In contrast and only in non-diabetic adults, vitamin E (800 IU α-tocopherol daily) can improve liver enzymes and histologic pathology [11,15].

The Mediterranean Diet, which is characterized by a high content of antioxidants and fiber, a balanced lipid profile and a low content of simple sugar, seems to be the optimal diet for the management of NAFLD. This diet is a natural multi-ingredient supplement that may exert its related health benefits by the synergistic and/or complementary action of each food compound [16]. A Mediterranean diet low in carbohydrates mobilizes more liver fat compared to a low fat diet with a similar weight loss [17].

### 2.2. Alcoholic Steatohepatitis (ASH), Liver Cirrhosis, and Acute Liver Failure

The nutritional recommendations for patients with ASH and liver cirrhosis are fundamentally different compared to the recommendations for patients with NAFLD, because these patients have a high risk for protein depletion, trace elements deficiency, and malnutrition. Twenty percent of patients with a compensated and 60% of patients with a decompensated liver disease are malnourished [18,19,20].

Therefore, screening for malnutrition is highly recommended on a regular basis in patients with ASH and liver cirrhosis, but the nutritional assessment can be difficult in patients with cirrhosis especially if there is associated fluid retention and/or obesity. Patients with cirrhosis may have a combination of loss of skeletal muscle and gain of adipose tissue, culminating in the condition of “sarcopenic obesity”. In addition, patients had a loss or deficiency of several other nutrients such as vitamin D and zinc [21]. The Nutritional Risk Screening-2002 and the Malnutrition Universal Screening Tool are validated and well-known tools to screen patients at risk for malnutrition [22]. Specifically in patients with liver disease, the Royal Free Hospital Nutrition Prioritizing Tool (RFH-NPT) was developed and when compared to Nutritional Risk Screening, it was more sensitive to identify malnourished liver patients. Next to the important variables unplanned weight loss and reduced dietary intake, the RFH-NPT has additional score points for complications like fluid overload and diuretics [23,24]. In addition, the presence or absence of sarcopenia may be assessed with radiological methods, because sarcopenia is a strong predictor of mortality and morbidity in patients with liver disease [25,26]. Next to radiological methods, handgrip strength is a simple, objective, and practical method to assess sarcopenia [21]. 

In case of malnutrition, patients need extensive nutritional counseling and therapy. The treatment is a challenge as the nutritional problems are multifactorial [11]. Usually, the resting energy expenditure is increased [27,28] and patients are required to consume 35–40 kcal/kg body weight [23]. Non-malnourished patients should eat 1.2 g protein/kg body weight per day to cover the protein needs, whereas the optimal intake of malnourished and/or sarcopenic patients is 1.5 g protein/kg body weight [11,23]. If energy and protein intake are not adequate with regular meals, oral nutritional supplements (ONS), or artificial nutrition (tube feeding or parenteral) should be used to avoid or treat malnutrition. However, the oral route should be preferred over enteral or parenteral nutrition. The standard formulas should be used, preferably formulas with high energy density (≥1.5 kcal/mL). In case of tube feeding, a percutaneous endoscopic gastrostomy placement is not recommended and can only be used in exceptional cases because of a higher risk of complications, e.g., infections, ascites, or oesophageal varices [22].

Patients with ASH or liver cirrhosis have poor hepatic glycogen stores due to the impaired synthetic capacity of hepatic cells, hence, an overnight fast in these patients is equivalent to a nearly 72 hours fast in healthy persons. As a result, metabolism shifts to fatty acids as a dominant substrate for oxidation. Some tissues dependent on glucose will need neoglucogenesis from amino acids as fatty acids cannot be used for this process. This leads to mobilization of amino acids from the skeletal muscles so that the adequate amount of glucose can be produced. Repeated and frequent fasting results in recurrent proteolysis resulting in muscle loss in human cirrhotic patients [21,29]. Therefore, fasting periods should be avoided and patients with severe liver disease, who have to fast for more than twelve hours, should receive intravenous glucose (2–3 g/kg body weight). When the fasting period lasts longer than 72 hours, total parenteral nutrition may be required [11]. At home, periods of starvation should also be kept short and, therefore, the consumption of three to five meals a day and a late evening snack are recommended [30]. One complication of severe liver disease is hepatic encephalopathy, a disorder of the central nervous system with a wide spectrum, ranging from psychomotor impairments to coma [31]. In case of hepatic encephalopathy, protein intake should no longer be restricted in cirrhotic patients as it increases protein catabolism and thus promotes malnutrition [32]. Advanced cirrhosis patients could benefit from branched-chain amino acids (0.25 g/kg body weight) in order to improve event-free survival and quality of life [33]. 

Acute liver failure may occur as a medical emergency. In multi-organ failure, a severe derangement of the whole metabolism can occur in these patients due to loss of hepatocellular functions. The metabolic condition is characterized by impaired hepatic glucose production and lactate clearance as well as protein catabolism associated with hyperaminoacidemia and hyperammonemia [11,34,35]. In general, patients with acute liver failure should be treated in the same way as other critically ill patients. Therefore, the patients need to be monitored regularly and if necessary, macronutrients, vitamins, and trace elements should be supplemented [11]. On the one hand, the energy requirement of patients with acute liver disease is increased [36], but on the other hand, a hypercaloric diet induces hyperglycemia and hyperlipidemia. Therefore, it is important to achieve an iso-energetic diet in order to avoid malnutrition and complications. For metabolic monitoring, the following target values should be aimed at: Blood glucose 8–10 mmol/L, serum lactate <5 mmol/L, triglycerides <3 mmol/L, and ammonia <100 mmol/L. The protein requirement is set at 0.8–1.2 g/kg body weight [11]. A summary of nutritional recommendations in liver disease are presented in Figure 1. 

## 3. Pancreatic Diseases

### 3.1. Acute Pancreatitis

The degree of the inflammatory response of the pancreas plays an important role for the assessment of clinical nutrition. Normally, patients with a mild or moderately severe pancreatitis do not need a specific nutritional? intervention, but every patient with acute pancreatitis should be screened for malnutrition with the regular tools and in case of malnutrition, receive an adequate nutritional therapy [37]. The energy and protein requirements are usually not increased and the patients are allowed to eat normal food independently of lipase and amylase activity [37,38]. In contrast to former medical opinions, fasting after an inflammatory episode has no positive effect on the clinical course or prognosis of acute pancreatitis. During the course of pancreatitis, exocrine secretion is blocked and therefore a stimulation of the exocrine function by food intake or artificial nutrition is not expected [6,37]. For those patients who can tolerate an oral diet, an initial low-fat solid diet is preferred [39,40]. This early approach to oral feeding may reduce the length of hospital stay in these patients [41].

In contrast, patients with severe necrotizing pancreatitis need an adequate clinical nutritional? strategy. Initial short-time fasting could be beneficial for patients with ileus or nausea and vomiting, but within 24–48 hours, enteral nutrition should be started. Early enteral nutrition has a more preventive then a nutritive effect, because early enteral nutrition decreased mortality and complications [42,43,44]. Parenteral nutrition instead could result in an intestinal villous atrophy within a few days, which then facilitates bacterial translocation and may result in severe infections. The administration of enteral nutrition counteracts translocation, and therefore enteral nutrition should be administered whenever possible to prevent intestinal atrophy [6,45].

Due to inflammation and pain, patients with severe acute pancreatitis have an increased energy and protein requirement [46]. Energy requirement is set at 25–30 kcal/kg body weight, glucose intake at 2–4 g/kg body weight, and protein at 1.2–1.5 g/kg body weight. Infusion rate for lipids should be 0.8–1.5 g/kg body weight and the infusion should be monitored regularly since triglycerides in plasma should be <12 mmol/L. Trace elements are supplemented in normal concentrations and high-dose vitamin supplements are not required [37]. In addition, administration of pre- and probiotics as well as immunonutrition cannot be recommended since studies so far have not shown a clear beneficial effect [47,48]. To reach the nutritional goal, enteral nutrition can be completed with oral food intake or parenteral nutrition if necessary. In case of total parenteral nutrition over a long time period, administration of 0.2–0.5 g glutamine/kg body weight could protect against infections and reduce mortality [49]. In addition, monitoring of volume substitution is very important and can decrease mortality [6]. An early return to normal food intake should be pursued [6]. After an episode of acute pancreatitis, approximately 20% of the patients develop the common complications diabetes and exocrine pancreatic insufficiency [50]. Therefore, patients should be regularly screened for these complications in order to prevent nutritional status.

### 3.2. Chronic Pancreatitis

Patients with chronic pancreatitis have a high risk for malnutrition due to maldigestion from pancreatic exocrine insufficiency in combination with inflammation and increased energy metabolism. Pancreatic exocrine insufficiency is the main pancreatic cause of malnutrition in these patients [51]. In addition, due to diarrhea and steatorrhea, patients with chronic pancreatitis often have a deficiency of the fat-soluble vitamins A, D, E, and K. Malnutrition is associated with an increased complication rate and increased mortality [52]. Therefore, malnutrition should be avoided using nutritional counseling and if necessary artificial nutrition. Energy requirement is set at 25–30 kcal/kg body weight and protein requirement at 1.5 g protein/kg body weight. Since alcohol is an important cause for chronic pancreatitis, patients should avoid alcohol completely [6,37,53].

In the case of pancreatic exocrine insufficiency, patients should be supplemented with pancreas enzymes. The exocrine insufficiency is diagnosed if the patients have either steatorrhea (>15 g/day fecal fats) or suffer from manifest maldigestion, respective malabsorption. For a first indication, patients can be asked whether they see undigested food in their feces or if the feces are difficult to wash away. Further examination could be a pancreatic function test and the evaluation of maldigestion-related symptoms like diarrhea or flatulence, poor nutritional status, and fecal elastase-1 concentration [51,52]. In patients with an exocrine insufficiency, low levels of circulation fat-soluble vitamins, proteins like albumin, lipoproteins, apolipoproteins, and mineral trace elements like magnesium, zinc, or calcium, can occur [54,55,56,57]. Since most of these abnormalities are related to pancreatic exocrine insufficiency, a laboratory analysis might be helpful in order to diagnose the insufficiency. However, other factors like toxic habits or deficient food intake may play a relevant role, too [51]. Pancreas enzymes should be taken during the meals and the dosage is based on lipase activity. A range of 20.000 to 40.000 lipase units per main meal should be administered as an initial dose and 10.000 to 20.000 units for the digestion of smaller in-between snacks [6,58]. It is not necessary to avoid high fat intake if the exocrine pancreas function is compensated [37]. The replacement therapy should promote digestion, but a complete normalization of digestion is usually not achieved. However, there are several options for a failure of normalization. First, the pH value in the stomach inactivates the pancreatic enzymes. Therefore, the addition of a proton-pump inhibitor before breakfast and dinner is recommended in cases of an unsatisfactory clinical response to the standard dose of pancreatic enzymes [59]. Second, the enzyme dose should be increased if needed to normalize digestion and the nutritional status of the patients. If these two strategies fail, another cause for maldigestion like bacterial overgrowth should be evaluated [51]. A summary of nutritional recommendations in pancreatic disease is shown in Figure 2.

## 4. Inflammatory Bowel Diseases (IBD)

IBD is a heterogeneous and multifactorial disorder resulting from a complex interplay between genetic variation, intestinal microbiota, the host immune system, and environmental factors such as diet, drugs, breastfeeding, and smoking [60,61,62]. The relationship between dietary nutrients and intestinal homeostasis is complex and influenced by several interactions between host immune system, the intestinal barrier, and the gut microbiota [60,63]. Patients with IBD have a high risk for malnutrition, which may be the result of reduced oral intake, increased nutrient requirement, increased gastrointestinal losses of nutrients, and occasionally from drug–nutrient interactions [7,64]. In pediatric patients, malnutrition is the main cause for growth retardation [65,66]. In CD, malnutrition is a great problem compared to UC because any part of the gastrointestinal tract can be affected. Therefore, the risk for malnutrition remains even when the disease is quiescent. UC is normally restricted to the colon and hence shows few malabsorptive problems except in active disease [64]. Due to the high risk, patients with IBD should be screened for malnutrition using the established tools at the time of diagnosis and thereafter on a regular basis [67,68]. Malnourished patients should receive an adequate nutritional therapy because otherwise it worsens the prognosis, complication rates, mortality, and quality of life [67,69,70]. The energy requirement of patients with IBD are normally not increased as well as the protein requirement in remission. In contrast, protein requirement is increased in active IBD and therefore, the protein intake should be 1.2–1.5 g/kg body weight [67,71,72]. 

In addition, patients with IBD have a high risk for micronutrient deficiencies due to losses from diarrhea and/or inadequate dietary intake. The most common micronutrient deficiencies are iron, calcium, selenium, zinc, and magnesium depletion. Vitamin deficiencies include all vitamins and in particular B_12_, folic acid, and vitamins A, D, and K [73,74]. For example, selenium, zinc, and magnesium depletions are caused by an inadequate dietary intake and chronic loss because of diarrhea. Symptoms associated with deficiencies include bone health impairment, fatigue, poor wound healing, and cartilage degeneration [73,74]. An example for the influence of medication is cholestyramine that can interfere with absorption of fat-soluble vitamins, iron, and B_12_ vitamin. Main side effect is steatorrhea due to impair absorption of fats [75]. Therefore, laboratory values of patients should be checked on a regular basis and possible deficits should be appropriately corrected. 

The most frequent extraintestinal manifestation of IBD is iron-deficiency and anemia, which occurs more frequently in CD and which should be supplemented with iron. Anemia is usually associated with other important symptoms like fatigue, sleeping disorders, restless legs syndrome, or attention deficit [75]. Patients with mild anemia can receive oral iron, when they are tolerant for oral iron and when the disease is inactive. Intravenous iron should be considered in patients with active IBD, with previous intolerance to oral iron, with hemoglobin below 100 g/L, and in patients who need erythropoiesis-stimulating agents [67,76,77]. Furthermore, patients may have deficiencies of calcium, vitamin D, folate, vitamin B_12_, and zinc [64]. When more than 20 cm of the distal ileum is resected, vitamin B_12_ must be administered to patients [67].

Low levels of calcium and vitamin D are common in patients with IBD, especially in those with duodenal and jejunal disease [61,73,74]. Calcium deficiency is linked to vitamin D deficiency, which is related to inadequate daily intake, inflammation status, diarrhea, and glucocorticoid therapy. The prevalence among IBD patients is up to 70% in CD patients and up to 40% in UC patients. Nonetheless, it is not established if the vitamin D deficiency is a cause or a consequence of IBD. However, there are suggestions that in genetically predisposed individuals, vitamin D deficiency may be a contributing factor in the development of IBD [78]. Beneficial effects of vitamin D in IBD are supported by pre-clinical studies mainly in mouse models, where the active form of vitamin D has shown to regulate gastrointestinal microbiota function and promote anti-inflammatory response [61]. 

When oral nutritional intake is insufficient during active disease, ONS are the first step. If oral feeding is not sufficient, tube feeding is superior to parenteral feeding. Parenteral nutrition is indicated in IBD (i) when oral or tube feeding is not sufficiently possible, (ii) when there is an obstructed bowel, where there is no possibility of placement of a feeding tube beyond the obstruction or where this has failed, or (iii) when other complications occur such as an anastomotic leakage or a high-output intestinal fistula [67]. During active disease, specific formulations or substrates, e.g., glutamine, ω-3 fatty acids are not recommended, neither is the use of probiotics. Probiotic therapy using Escherichia coli Nissle 1917 or VSL#3 can be considered for use in patients with mild to moderate UC for the induction of remission [79]. 

Enteral exclusive nutrition has been extensively used for induction of remission in pediatric CD, in which avoidance of steroids is critical for childhood growth. Several recent pediatric studies have demonstrated and confirmed that enteral exclusive nutrition can induce remission in 60–86% of children [80,81,82] and is associated with higher remission rates, better growth, and longer steroids-free periods [60]. However, the benefit was lost when partial enteral nutrition was used with access to a free diet [83]. Nonetheless, a recent study has shown that partial enteral nutrition can be effective for induction of remission in children and young adults in combination with a diet, which is based on components hypothesized to affect the microbiome or intestinal permeability [84]. In remission, ONS or artificial nutrition are only recommended if malnutrition cannot be treated sufficiently by dietary counseling. In addition, specific diets or supplementations with ω-3 fatty acids are not recommended for maintenance of remission. A systematic review has not supported the hypothesis that supplementation of ω-3 fatty acids can induce and maintain remission in IBD [85]. However, several studies have demonstrated that different genotypes can be associated with the variable response to nutritional intervention with ω-3 fatty acids [86]. Probiotic therapy can be considered in UC but not CD for maintenance of remission [67]. In addition, clinical trials show that curcumin supplementation might be effective for the induction and maintenance of remission in UC patients [87,88]. Curcumin suppresses cytokine production by macrophages and intestinal epithelial cells via the inhibition of NF-kB activation [89,90] and thus mitigates induced colitis in mouse models [91,92]. A summary of nutritional recommendations in IBD is presented in Figure 3.

## 5. Future Perspectives

Clinical nutrition is clearly important in the treatment of patients with liver, pancreatic, and inflammatory bowel disease. However, there are many gaps of knowledge in the pathophysiology of these diseases despite an enormous research effort and thus the effects from clinical nutrition cannot be at a maximum so far. 

A major gap in knowledge is associated with the evolution of NAFLD in children and adolescents. Weight loss is important for histological improvement and the benefits of weight loss will extend beyond those expected from drug treatment of high-risk NASH. It is postulated that only public health strategies will have the opportunity to improve the burden of obesity-related diseases in the future [93]. In addition, due to our aging population, it is expected that the burden of liver disease due to cirrhosis from NASH will increase over the next two decades unless effective preventive and therapeutic interventions are implemented as part of a public health strategy [93].

Furthermore, knowledge about the human microbiome constantly increases. Trillions of microbial cells, which form a symbiotic relationship with the host, play a crucial role in the development of diseases, when the balance of the microbiome becomes disrupted [94]. Increase intestinal permeability and dysbiosis are common characteristics linking the liver to a number of gastrointestinal diseases [95]. For example, alcohol consumption and endogenous alcohol production by gut bacteria in obese individuals can disrupt the tight junctions of the intestinal epithelial barrier, resulting in increased gut permeability. The bacterial endotoxin in the portal circulation leads to liver inflammation and fibrosis through activation of toll-like receptor 4 [95]. Although several mechanisms by which the microbiome might affect liver disease have been proposed, more work is needed to fully understand these relationship [93].

IBD is also associated with a dysbiotic microbiome. However, it is unclear if this dysbiosis plays a role in the pathophysiology or is a result of the disease. Due to the importance of the microbiome in IBD, therapies manipulating the microbiome have gained popularity. While there are some promising trials demonstrating the efficacy of antibiotic combinations in treating IBD, more controlled trials are required [94]. In addition, a more precise understanding of the complex interrelation between dietary nutrients, host immunity, and the microbiome is necessary to increase the effectiveness of dietary interventions used to treat IBD [63]. Moreover, an in-depth knowledge of the genetic background is important for personalized nutritional management, which might lead to a maximum efficacy in therapy [63].

New trials in acute pancreatitis showed that approximately one-third of patients will develop prediabetes or diabetes within five years of an index episode of acute pancreatitis and 24–40% of the patients develop an exocrine pancreatic insufficiency, but the mechanisms and risk factors remain to be specified [96,97,98,99]. Furthermore, further evidence is required to determine the optimal panel of laboratory markers for nutritional evaluation in chronic pancreatitis, and the utility, reliability, and accuracy of these markers in diagnosing pancreatic exocrine insufficiency [51].

## Figures and Tables

**Figure 1 jcm-08-01098-f001:**
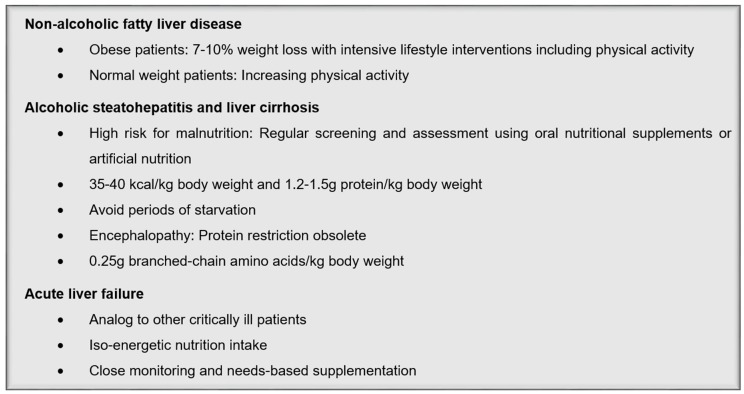
Summary of nutritional recommendations in liver disease.

**Figure 2 jcm-08-01098-f002:**
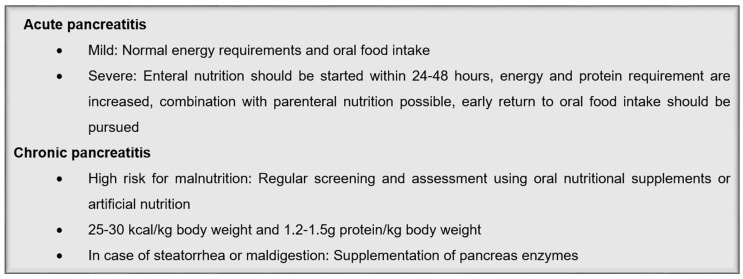
Summary of nutritional recommendations in pancreatic disease.

**Figure 3 jcm-08-01098-f003:**
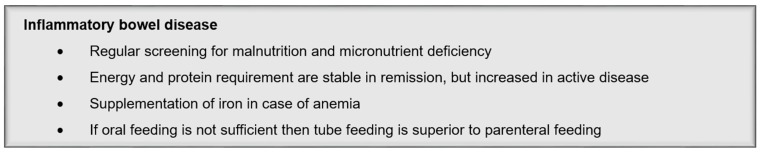
Summary of nutritional recommendations in inflammatory bowel disease.

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
