# Peer review of "Nutrition in Gastrointestinal Disease: Liver, Pancreatic, and Inflammatory Bowel Disease"

_jcm, 2019, doi:10.3390/jcm8081098_

Round 1

Reviewer 1 Report

The review by Storck and colleagues focuses on the nutrition in disorders of liver, pancreas and the bowel. The review is well-written and contains all of the basic information necessary in the daily clinical practice including dosing of nutrients and biochemical markers used for monitoring of patients’ condition. The literature cited in this paper is up-to-date. I have the following general, major and minor comments:

General: I have read this review with interest however going through the whole manuscript left me unsatisfied due to its low novelty. This article is more of the handbook chapter for gastroenterologist, dietician or perhaps patient. I would like authors to include more recent findings, perhaps even experimental ones, to increase the scientific value and make the article more exciting to the reader.

Major issues: Given the comment above I recommend adding a paragraph about future perspectives in nutrition in liver, pancreas and bowel diseases.

I see a striking imbalance between information covering the nutrition in CD vs. UC. Authors should elaborate more on UC and its potential implications in nutrition status of the patient as well as potential dietary interventions.

Minor issues: an abbreviation for NASH should be given at the first use in the manuscript – line 95, p. 3

Delete “result” from the phrase “...should result promote…” – line 210, p.8

Author Response

Thank you very much for your recommendations. We tried to implement all of your suggestions and added a part of future perspectives to the conclusion part. In addition, we added new information for all diseases.

Reviewer 2 Report

The authors have done a quite satisfactory effort to provide the reader with nutritional guidelines concerning a number of liver, and pancreatic disorders as well as to give some directions regarding the diet in inflammatory bowel disease patients.

However, there is an abundance of relevant references dealing with the subject, in which the readers could find all the information required.

Examples are the following:

LIVER DISORDERS

Ullah R, Rauf N, Nabi G, Ullah H, Shen Y, Zhou YD, Fu J. Role of Nutrition in the Pathogenesis and Prevention of Non-alcoholic Fatty Liver Disease: Recent Updates. Int J Biol Sci. 2019;15:265-276.

Yamazaki T, Li D, Ikaga R. Effective Food Ingredients for Fatty Liver: Soy Protein β-Conglycinin and Fish Oil. Int J Mol Sci. 2018;19(12).

Mandato C, et al.Nutrition and Liver Disease.Nutrients. 2017;10(1).

PANCREATIC DISORDERS

Dominguez-Munoz JE, et al. Recommendations from the United European Gastroenterology evidence-based guidelines for the diagnosis and therapy of chronic pancreatitis. Pancreatology. 2018;18:847-854.

Löhr JM, et al. United European Gastroenterology evidence-based guidelines for the diagnosis and therapy of chronic pancreatitis (HaPanEU). United European Gastroenterol J. 2017;5:153-199.

IBD

Scaldaferri F, et al. Nutrition and IBD: Malnutrition and/or Sarcopenia? A Practical Guide. Gastroenterol Res Pract. 2017;2017:8646495.

Halmos EP, Gibson PR.Dietary management of IBD--insights and advice.Nat Rev Gastroenterol Hepatol. 2015;12:133-46.

Eder P, et al.Dietary Support in Elderly Patients with Inflammatory Bowel Disease. Nutrients. 2019;11(6).

Fletcher J, et al.The Role of Vitamin D in Inflammatory Bowel Disease: Mechanism to Management. Nutrients. 2019;11(5). 

Author Response

Thank you very much for your suggestions. We included almost all of your suggested references and thus updated information about all three diseases.

Reviewer 3 Report

This is a short review that focus on the clinical nutritional practice in liver diseases, pancreatitis and inflammatory bowel disease. Even so, the article should become more interesting if the authors choose to go deeper into some of the issues: 

Liver: The introduction to the metabolic functions of the liver is very brief. Would a table with the metabolic functions of the liver and how they are changed by caloric overload, alcohol, portal shunting, etc be appropriate? Mechanisms behind the muscle mass loss in cirrhosis would be of particular interest (See e.g. Anand 2017; J Clin Exp Hepatol 7:340-357). There are many recent reviews that deal with the complex interactions between diet, microbiota etc in the progression of simple fatty liver to NASH, cirrhosis, primary liver cancer. A couple of these could be referred to. The evidence summarizing efefcts of a Mediterranian style diet and potentially effective additions as vitamin E, other antioxidants, choline, n-3 fatty acids has also been reviewed and could be referred to (e.g. Suarez et  al Nutrients 2017; 9:1052). There is a recent update on Espen recommendations on nutrition in liver disease (Plauth et al Clin Nutr 2019; 38:485-521) that could be referred to.

Pancreas: Lee and Papachistou Nat Rev Gastroenterol may 2019; New insights in acute pancreatitis, also give secondary references to nutrition.

Inflammatory bowel disease: This part could be extended. It does not comment on pediatric IBD at all. Potential role of vitamin D and other micronutirints could be treated in more depth. The complex issue whether diet may influence course of the dises and risk of relaps is ciurrently receiving increasing interest (e.g. Sáez-Gonzales et al; Bases for nutritional recommendations for patients with inflamamtory bowel disease, Nutrients 2019, and Sugihara et al Front Immunol 2019). 

Author Response

Thank you very much for your suggestions. We added information about the mechanism behind the muscle mass loss in cirrhosis, about the liver-gut-axis, about Mediterranean Diet and further micronutrients in liver disease. We included information from the work of Lee and Papachistou for pancreatic disease and we extended the part for IBD including pediatric IBD, vitamin D and further micronutrients as well as the complex issue of the course of disease. The only advice that we did not realize, was the table with metabolic liver functions. We already see an imbalance between the part about liver disease compared to pancreatic disease and IBD. Therefore, we decided against this table.

Round 2

Reviewer 1 Report

The manuscript has been improved. My comments were addressed adequately. I have no further comments.

Reviewer 2 Report

There are no further suggestions. The paper has been improved.

Reviewer 3 Report

The authors have considered most of my earlier criticisms, and the paper may now be valuable reading for people who are not specialized in nturition but who want to broaden their nutritional knowledge in an area whre this is indeed important. Some linguistic details can be improved, but the paper can now be published after such a final workover.